# Navigating Nutrition Beyond Elite Sport: A Qualitative Exploration of Experiences After Retirement

**DOI:** 10.3390/nu17243920

**Published:** 2025-12-15

**Authors:** Ebeney K. Whillas, Joel C. Craddock, Kelly Lambert

**Affiliations:** School of Medical, Indigenous and Health Sciences, University of Wollongong, Wollongong, NSW 2500, Australia; ekw983@uowmail.edu.au (E.K.W.); jcraddock@uow.edu.au (J.C.C.)

**Keywords:** athlete, qualitative, eating, sport, disordered eating

## Abstract

**Background/Objectives**: Retirement from elite sport often disrupts structured routines and performance-driven nutrition habits, leaving athletes vulnerable to maladaptive eating behaviours and body image concerns. This study aimed to explore the experiences of former elite athletes regarding healthy eating after retirement, focusing on preparedness, barriers, and enablers during the transition to post-sport life. **Methods**: A qualitative design was employed using semi-structured interviews with former Australian athletes (national, international, or Olympic level) recruited via snowball sampling and professional networks. Interviews were recorded, transcribed, and analysed using an inductive thematic analysis framework to identify key themes and subthemes. **Results**: Sixteen elite or highly trained athletes (56% female) were interviewed. Four overarching themes were apparent: (1) navigating life beyond elite sport, (2) detaching from sporting culture and belief systems, (3) reframing food, body, and control, and (4) the journey to healthy behaviours and food freedom. Participants reported identity loss, inadequate transition support, and persistent body image concerns. Over time, many described a gradual shift towards intuitive eating and improved relationships with food and self, though residual “food noise” and restrictive tendencies persisted for some. **Conclusions**: The findings highlight the need for athlete-centred dietetic and psychological interventions across the athletic lifecycle and post-retirement. Culture change within elite sport and the development of tailored, accessible transition resources that include digital and AI-supported tools may facilitate healthier eating behaviours and long-term wellbeing.

## 1. Introduction

Elite athletes maintain highly structured routines and performance-driven nutrition throughout their careers. However, retirement from elite sport represents a major life transition that can disrupt these routines and challenge physical, psychological, and social wellbeing. Evidence suggests that this transition often leads to identity loss, reduced physical activity, and increased vulnerability to maladaptive eating behaviours such as restrictive eating and compensatory changes to exercise, as well as body dissatisfaction and lifestyle-related chronic disease risk [1,2,3,4,5,6].

Sporting culture exerts a strong influence on athletes’ beliefs about food and body image. Years of exposure to weight-focused coaching, rigid dietary practices, and performance-based nutrition can leave lasting imprints that persist beyond retirement [2,7,8]. Former athletes frequently report difficulties adjusting to unstructured eating patterns, managing weight fluctuations, and overcoming deeply ingrained attitudes towards food. Studies indicate that retired athletes experience higher rates of disordered eating compared to the general population, particularly those from aesthetic or weight-dependent sports [6].

Despite growing recognition of post-retirement challenges [2], the available supports remain limited and often prioritise career planning over nutrition and wellbeing after retirement [9]. In addition, much of the existing evidence comes from women in weight-centred sports, meaning that male athletes from non-weight-dependent sports are comparatively underrepresented. Given the potential longer-term health implications, including increased cardiovascular risk, there is a need to better understand how retired athletes conceptualise and practice healthy eating after retirement, and whether cultural beliefs from sport shape their ongoing relationship with food. By using a qualitative approach, this research examines the preparedness of athletes to adopt healthy food choices and identifies factors influencing dietary behaviours during the transition to life beyond sport.

## 2. Materials and Methods

This study utilised a qualitative research approach embedded in a relativist ontological lens [10]. Ethics approval was obtained by the Human Research Ethics Committee through the University of Wollongong and all participants provided informed consent prior to participation. The inclusion criteria included former athletes now aged 18 or more years who had competed at a national, international, and/or Olympic level in any sports. Specialised and high-level Defence Force personnel, such as special forces and high-ranking trainers, were also eligible to be included due to their vigorous, regimented dietary intake routines and highly structured training requirements similar to that of elite and highly trained athletes. These individuals operate under externally imposed control of food intake and may experience identity disruption when leaving the service, paralleling the psychological and behavioural challenges that may be faced by retired athletes. Their inclusion allowed for an exploration of whether these shared cultural and structural dynamics produce similar post-transition experiences.

Athletes were recruited via snowball sampling, with initial posts made on social media platforms such as Instagram, LinkedIn, and Facebook. Other recruitment methods included distribution of the advertising material to professional sporting networks of the research team as well as organisations such as Sports Dietitians Australia.

Potential participants were encouraged to complete an expression of interest form with contact details and the sport they competed in prior to participating. Each athlete was then contacted individually to sign consent forms and schedule an interview. Interviews were conducted via Zoom or in person by the author and were audio-recorded, and the audio recordings were uploaded to Otter AI (AISense, 2016, Mountain View California) software for transcription. The transcripts were cross-checked against the recording for accuracy and deidentification.

Regarding reflexivity, snowball sampling and reliance on personal networks were chosen to access a hard-to-reach population of retired elite athletes. While these strategies may facilitate recruitment, they may also shape the composition of the sample by favouring individuals who maintain connections to sporting communities or feel comfortable sharing their experiences. This could potentially amplify certain narratives, such as those with strong opinions about transition challenges, and risks obtaining limited perspectives from athletes who have disengaged completely from sport. We acknowledge this as a potential source of selection bias and suggest interpreting the findings with this context in mind.

The semi-structured interview guide (Table 1) comprised six demographic questions and ten questions exploring dietary intake and eating patterns prior to and after retirement with a specific focus on the influence on food choices, eating behaviours, and access to support. Questions were developed by the research team based on clinical experience and recent data released on athletes who have higher rates of coronary atherosclerosis compared to controls [11]. The participants could cease interviews at any time.

We used the following constructs in this study to help delineate subjective meanings from normative assumptions and to enable readers to interpret participant accounts. Healthy eating refers to a flexible, balanced approach to food that supports both physical and psychological wellbeing in everyday life, rather than a regime optimised for competitive performance. We use food freedom to refer to autonomy in food choices without rigid control or guilt, acknowledging that freedom does not imply unlimited intake but brings reduced cognitive burden and increased trust in internal cues [12]. A nutritious diet refers to a dietary pattern that meets physiological needs for long-term health, distinct from sport-specific periodisation or physique-driven targets [1].

This research utilised inductive thematic analysis and follows the Braun and Clarke 6-step thematic analysis framework [13]. Initially, each interview was recorded and uploaded to Otter AI (AISense, 2016, Mountain View California) and transcribed. Each interview transcript was uploaded to Dedoose data analysis software for initial coding. Initial codes were generated inductively by the first author. Interpretive rigour was maintained by involving all members of the research team in the analytic process. A sample of 10% of the transcripts were coded in triplicate to determine alignment. Initial in vivo coding consisted of reading each transcript and highlighting any key words, quotes, or phrases that were of interest. After all interviews were coded, an Excel spreadsheet of all codes was downloaded and used to collate each code into potential categories, which comprised reoccurring words, phrases, or topics. These categories were then grouped together in meaningful ways and labelled as potential themes or subthemes. Once these were collated, collaborative review sessions were undertaken weekly where members compared interpretations, assumptions, and alternative wordings or framing. The themes were thus revised iteratively. These discussions were also informed by our professional backgrounds in nutrition and dietetics, which helped surface implicit biases around concepts such as “healthy eating” and “food freedom”. A thematic map was developed to explore relationships between concepts and assist with further refinement. To ensure validity, the participants were offered the opportunity to review the transcripts and final themes, but this was declined by all participants. To maintain reflexivity, weekly peer debriefing occurred with members of the research team with critical self-reflection on each researchers’ role and potential biases.

Regarding potential participant distress, both senior researchers are qualified health professionals with extensive counselling experience and were available to provide debriefing if desired. In cases of severe distress, mental health first aid protocols were in place, including adopting an attitude of acceptance and empathy, active listening, and positive body language. The main interviewer (EW) maintained a reflexive diary to monitor emotional tone and her own responses. EW debriefed with KL weekly to reflect on participant wellbeing and interviewer influence. These measures were designed to ensure ethical reflexivity and emotional safety throughout the study.

## 3. Results

In total, 16 former athletes (females *n* = 9, males *n* = 7) who competed at a national, international, or Olympic level of sport volunteered their time and participated in this study. The characteristics of participants are shown in Table 2. Descriptive information about the level of elite sport competed in were self-reported by participants using a six-tiered Participant Classification Framework of athlete definitions [14] (see Appendix A) that provides specific athlete classifications that can be utilised in research relating to sport, performance, and health (13). In brief, of the participants, 56% (*n* = 9) were female. All participants (100%) had competed professionally at or before the age of 25 years old, and only 29% (*n* = 4) continued competing beyond the age of 25. Of the athletes, 36% (*n* = 5) had competed at level 3 (Highly trained); 57% (*n* = 8) at level 4 (Elite), and 7% (*n* = 1) at level 5 (World class). The age of participants during elite competition ranged from 5 to 64 years. Athletes competed in a range of team and individual sports including Ballet, Triathlon, Artistic Gymnastics, American Football, Baseball, Bodyboarding, and Cycling. Two Defence Force personnel participated who were involved in the Special Forces, as well as the Australian Services Rugby League.

Four themes were identified that characterise the experiences of healthy eating after retirement:Navigating life beyond elite sport;Detaching from sporting culture and belief systems;Reframing food, body, and control;The journey to healthy behaviours and food freedom.

These themes are interrelated and the underlying concepts included are sequential, and are depicted in Figure 1. The themes suggest that the internal reframing of the connections among food, body, and control is eventually achieved. These challenges and experiences were not unique to women in aesthetic sports and were consistent across varying contexts, sports, genders, and ages.

### 3.1. Theme 1: Navigating Life Beyond Sport

Retirement was often described as a period of shock and disorientation. Athletes reported a lack of structured transition support, leading to a perceived loss of identity and drastic changes in routine. Many felt abandoned by their sporting organisations, with some being told to simply “*find a hobby*” or “*find a casual job*” (Participant 5, Ballet). Even if support had been made available, some athletes were reluctant to access it due to concerns about information being shared with coaches or teammates.

#### 3.1.1. Transition Support and External Influences

Participants consistently described inadequate support for post-retirement dietary needs, for example, “*Once athletes leave elite competition they get forgotten about*” (Participant 13, Triathlon). Some sought advice from dietitians but found the guidance too generic, lacking relevance to their unique experiences as athletes. One participant noted that they were being given general population advice and said “*I don’t have the mindset of a general population client*” (Participant 11, Swimming). They also stated: “*[the advice] Like, it wasn’t as specific or tailored as when I was competing*” (Participant 11, Swimming). Others turned to podcasts, social media, and online resources to fill the gap, with self-directed information seeking a means to support the move to autonomy.

#### 3.1.2. Identity Loss

Athletes described a deep psychological struggle with identity loss after retiring from elite sport, regardless of how retirement occurred. This was often rooted in years of immersion in their sport from a young age. Their sense of self was closely tied to performance, discipline, and social networks within the sporting environment. One participant reflected, “*I feel like I am back at square one with my life and career*” (Participant 8, Ballet). For those in highly structured environments like the Defence Force, the transition was especially difficult: “*You don’t just break out of discipline like that*” (Participant 15, Defence). Disruption of identity is linked by participants to dietary uncertainty; that is, many feel they have lost control over the strict environments shaping food encounters.

#### 3.1.3. Routine and Environmental Influences

Retirement disrupted established routines and social circles, leading to changes in eating behaviours. However, deeply ingrained beliefs from sporting culture persisted. One athlete remarked, “*the smaller and the leaner you are, the faster you go, not necessarily, the healthier you are. But that was like a strong message, like strong message that was sent out for a long time So I think, like the pressures of, I guess, looking a certain way, not only for like, body image, but also for performance, definitely then had a very negative impact on, like, my relationship with food for a long time. And then even after I kind of realized how negative that was, or, like, how toxic that was for performance and just my overall health, it still probably took another two years to be able to be able to actually turn it around and really change those habits*” (Participant 14, Triathlon). Many described relearning what normal intake looks like—“*So you do it [eat for performance] for 10 years, and then you go, how does a normal person eat? I have to unwind the last 10 years of my life?*” (Participant 1, American Football)—and discovering that food could serve purposes beyond performance, such as enjoyment, social connection, and celebration. The battle to learn “normal” is a pivotal transition point from externally controlled motives for eating to internally referenced motivations to eat.

### 3.2. Theme 2: Detaching from Sporting Culture and Belief Systems

Intentional detachment from the beliefs and norms of elite sport was a necessary but challenging process described by many. This was often due to misalignment with the athlete’s new lifestyle and perceptions of a healthy diet. Athletes described internal conflicts between deeply ingrained performance-driven food choices and emerging values around health and wellbeing.

#### 3.2.1. Psychological/Emotional Relationship with Food

Food was often viewed transactionally, that is, earned through effort or discipline. Many participants shared the perspective that food was fuel, not enjoyment, during competition, while another said food choices were related to “*what I felt I deserved*” (Participant 6, Ballet). This mindset persisted post-retirement, with food still perceived as a reward or something to be restricted unless “earned”. For example, one ex ballerina stated “*So, when you’re a ballet dancer, and if you get given like a role where you’re dancing a lot, I’d be encouraged to [eat] … it’s related to kind of feeling worthy, right? Like I am allowed to eat good quality food now*” (Participant 6, Ballet). Another athlete demonstrated this mindset in the context of retirement, saying that they “adapted quickly to cutting down intake as food wasn’t as enjoyable when you had not earnt it” (Participant 13, Triathlon). This concept of “deservingness” reflects moral schemas learned in elite sport, and post retirement appear as guilt, self-restriction, and maladaptive eating patterns.

#### 3.2.2. Adaptation and Mindset Shift

Time away from sport was essential for athletes to begin embracing healthier eating. One participant reflected, “*I hated anything to do with it (sport related activities)*” (Participant 3, Artistic Gymnastics), while another noted the risk of relapse into disordered behaviours when focusing on food too soon. The shift required a readiness to change, as one athlete put it: “*You can lead a horse to water, but you can’t make it drink*” (Participant 6, Ballet). This participant later mentioned that “*even now, I’ll have like moments, like little glimpses, little whispers of perhaps an old voice in my head with my relationship with food. But it comes so far and few between, and I can really see it so clearly for what it is, rather than letting it, you know, embody who I am*”, highlighting how time away from sport has supported a mindset shift, with gradual reframing beyond performance-focused values.

#### 3.2.3. Long-Term Impact of Sporting Culture

Sporting culture left lasting imprints on athletes’ attitudes towards food and body image. Those from aesthetic sports like swimming, dance, and gymnastics described experiences of weight bias, food demonisation, and restrained eating. These sports were described as perpetuating a focus on an ideal physique with severely negative impacts on body image perceptions, and major challenges related to deep-rooted beliefs that shape relationships with food beyond retirement. These beliefs shaped their post-retirement behaviours and contributed to ongoing struggles with body image and food-related guilt. Institutional practices and surveillance as well as the demonisation of foods and nutrients shape emotional responses to food and eating behaviours. Additional detail on how sporting culture impacts attitudes and behaviours is shown in Table 3.

### 3.3. Theme 3: Reframing Food, Body, and Control

Post-retirement eating was described by participants as requiring a conscious effort to reframe relationships with food, body, and control. This process was often marked by trial and error, especially for those with histories of disordered eating.

#### 3.3.1. Disordered Eating

Athletes described lingering restrictive behaviours and guilt around certain foods. One participant shared that “I had restricted myself for so long that I had a lot of food noise… I was constantly thinking about food [post-retirement]” (participant 7, Ballet). These intrusive thoughts were linked to fears of weight gain and deeply embedded beliefs about calorie control. As one athlete described, “*my mind just kind of flips and starts to go against me*” (Participant 8, Ballet), and many described their old mindset and beliefs continuing to impact their current intake, even if they were aware of it. Old mindsets around the need to restrict intake sometimes led to “end of day binges” (Participant 6, Ballet). This description of “food noise” reflects residual thoughts of chronic food restriction. The reported binges reinforce how rigid control can paradoxically destabilise healthy food intake and sustain guilt.

#### 3.3.2. Relationship with Self

Many participants described a journey to healthier, more balanced diets by healing their relationship with themselves. One participant mentioned that following a healthy diet post-retirement was “*absolutely unrelated to food or like, seeking out support around food. It’s 100% related to the relationship with myself*” (Participant 6, Ballet). Athletes described learning to appreciate the value of good food and the importance of fuelling their bodies. Social media played a role in normalising healthy eating and food behaviours and promoting positive self-image: “*ex Olympians obviously have a much greater platform because of social media, and there’s been a real change in attitudes and calling out unhealthy patterns* … *food is like a social thing, and can be like celebrated and you should enjoy it and it tastes good*” (Participant 11, Swimming).

#### 3.3.3. Body Image and Weight Stigma

Body image concerns were prevalent, particularly among women in aesthetic sports like swimming, ballet, and gymnastics. One participant noted: “*Intake was definitely not based around performance. It was about how I looked*” (Participant 5, Ballet). These perceptions were shaped by years of exposure to weight-focused coaching and peer comparisons, with one participant stating “*I think there’s a really big question mark around the coaches and the stuff that they’re saying kind of behind the scenes. So because those kinds of comments, they stick with you forever*” (Participant 11, Swimming). This stigma often followed into retirement, as another participant stated: “*I think I was actually a little bit worried because I wasn’t training at the same intensity, so I initially tried to restrict what I was eating even further than what I was already doing as a gymnast, and I would also up my training intensity level at a regular gym, not gymnastics, just to try to keep at a similar weight that I was and a similar body image. So I’d say probably was more restrictive*” (Participant 4, Artistic Gymnastics). These data illustrate how stigma acts as a mechanism to maintain control-based eating well into retirement.

### 3.4. Theme 4: The Journey to Healthy Behaviours and Food Freedom

The final theme reflects a gradual shift towards balanced eating and food autonomy. The athletes described overcoming barriers and learning to self-manage their intake in ways that supported overall wellbeing. The move to becoming more autonomous was uneven and iterative, often requiring new routines to be established and required a reframing of food’s purpose beyond performance (to reflect nourishment, connection, and pleasure).

#### 3.4.1. Barriers to Healthy Eating

Challenges included the lingering impact of sporting culture, financial constraints, time pressures, and a lack of tailored nutrition education. One athlete stated, “*I just think it takes a lot of athletes, a bit of time to kind of find that healthier relationship with food*” (Participant 14, Triathlon). Many participants turned to meal prepping, podcasts, and digital tools like Generative AI to support the transition to post-retirement healthy eating. Another barrier was whether or not the athlete felt prepared to adjust their intake: “*I didn’t’ [feel prepared] … The word for me is habit. I never formed a habit from a young age or through my adult life because I didn’t have to, you know … because I just trained all the time … I am finding it hard now, because it’s trying to, sounding very military, but discipline myself and my body to eat properly*” (Participant 16, Defence Force/Rugby League). The use of digital tools suggests emerging autonomy support strategies (to obtain timely information, engage in self-monitoring) at times when access to formal services might be unavailable or inaccessible.

#### 3.4.2. Learning to Self-Manage

The athletes described a sense of empowerment as they developed more intuitive and enjoyable eating habits. Participants described a shift in belief systems about food from being transactional to a more holistic view of food as a component of overall health and wellbeing. One participant shared that “*I can go out and just enjoy a meal with friends and have a drink and not be worried about, like, I guess, the impact that’s gonna have*” (Participant 14, Triathlon), and others spoke of communal meals and social eating as part of their healing journey. Others indicated that learning to self-manage their diet was a result of years of growth and unlearning deeply embedded behaviours, creating a sense of relief and empowerment for many participants. These moments mark the emergence of food freedom—that is, reduced cognitive load, rebuilding of confidence, greater trust in internal cues, and alignment with wellbeing goals.

#### 3.4.3. Residual Food-Related Thoughts

Despite progress, some athletes continued to experience intrusive thoughts shaped by their sporting past. One participant described the internal conflict between old habits and new knowledge: “*I can I still sometimes find myself slipping back into bad habits. … It’s athlete brain versus smart brain*” (Participant 11, Swimming). These residual thoughts highlight the enduring influence of elite sport on food-related behaviours and the nonlinear nature of change.

## 4. Discussion

The aim of this study was to explore the experiences of former elite athletes regarding healthy eating after retiring from elite sport. Four themes and twelve subthemes captured the complex psychological journey from the initial transition period to months or years later. These findings illustrate the enduring influence of sporting culture on eating behaviours, the lack of structured dietetic or psychological support post retirement, and the barriers and enablers that shape dietary practices and influence eating behaviours. Importantly, they reveal the gradual psychological process of developing a healthy relationship with food long after athletic careers end.

These findings align with existing literature on retired elite athletes, particularly female, highlighting challenges such as identity loss, body dissatisfaction, disordered eating, and the pervasive influence of sporting cultural norms [15]. A systematic literature review by Buckely et al. [1] reported that athletes often engage in maladaptive eating behaviours post retirement, including meal-skipping when exercise is missed, restrictive eating, and compensatory exercise to prevent perceived undesirable body composition change. The review also emphasised that an individual’s relationship with self strongly influences body acceptance and eating behaviours, a theme echoed in our own study. For example, one participant reflected that “*I initially tried to restrict what I was eating even further than what I was already doing as a gymnast, and I would also up my training intensity level at a regular gym … just to try to keep at a similar weight … and body image. So I’d say probably was more restrictive*” (Participant 4, Artistic Gymnastics). Another participant emphasised that healthy eating behaviours were “*100% related to the relationship with myself*” (Participant 6, Ballet). These narratives mirror previous findings that sustained athletic identity and body image ideals can perpetuate maladaptive eating behaviours during the transition to retirement [1,15].

The convergence between our findings and previous research suggests that disordered eating behaviours and psychological adjustment challenges are not isolated but represent recurring patterns across retired athlete cohorts. There is, therefore, an urgent need to implement tailored interventions to address the nutritional and psychological aspects of retirement. While some of our findings confirm existing evidence, we also offer several unique contributions. Much of the prior research has been conducted in highly weight-centric sports and has frequently focused on female athletes, often examining narrow samples or relying on secondary data sources [1,2,12,15,16,17]. As a result, male athletes and those from non-weight-dependent sports remain underrepresented [15]. In contrast, our study sample included men (44%, *n* = 7) and women (56%, *n* = 9) across nine athletic disciplines as well as two retired Defence Force personnel, a group rarely studied in this context. This diversity enables novel insights to be gained about how varied elite environments shape eating behaviours, identity, and health during the transition from sport [1,15].

Beyond confirming the known challenges, our study enables the identification of practical strategies and emerging opportunities to implement athlete-centred support. Several participants described using podcasts and generative AI tools like ChatGPT during the transition. These tools can operate as autonomy-supporting resources and connect users to non-judgemental support and information. Risks include accuracy, accountability, privacy, and overreliance without clinical oversight. These approaches are largely absent from the current literature, suggesting early signals of autonomy oriented self-care that may reduce pressure during the transition period [7]. Importantly, athletes emphasised the need for time away from sport before engaging with structured programmes, indicating that initial support should be external to sporting organisations to foster trust and uptake. Developing sport-specific digital resources, such as transition-focused podcasts or educational modules on safe and ethical AI use for nutrition, may bridge this gap until athletes are ready for formal services like tailored webinars, peer networks, and psychological counselling. This concept of self-directed digital support remains undocumented in prior literature and represents a promising research avenue as an enabler of athlete-centred dietetic and psychological care.

Unexpectedly, the four main themes and their sequential nature were consistent across gender, age, and sporting backgrounds. Both male and female athletes described lifestyle shifts and maladaptive eating behaviours during retirement, often while adjusting appetite and mindset. For instance, a ballet dancer accustomed to years of dietary restriction described their struggle to relinquish rigid control, whereas a former Defence Force member (previously unconcerned with intake) found monitoring food challenging as activity levels declined. Despite contrasting elite sporting environments, both individuals experienced heightened “food noise,” guilt, and ingrained beliefs about control, prompting a re-evaluation of food’s purpose. This consistency across diverse contexts reinforces the universality of these challenges and the need for comprehensive, individualised support.

The key strengths of this study are the inclusion of athletes from multiple sports, representation of both genders, and the novel addition of retired Defence Force personnel. The inclusion of Defence Force personnel was intentional, reflecting conceptual overlap with elite sport culture. Both contexts emphasise discipline, body composition monitoring, and performance-based nutrition, creating comparable challenges when structured routines end. While their narratives were analysed alongside athletes, we acknowledge this as a conceptual boundary and interpret the findings with caution. Future research could examine these groups separately to further delineate similarities and differences.

Unlike much of the existing literature, our qualitative design provides rich accounts of lived experiences before and after retirement. These perspectives are frequently absent from the existing evidence base. These insights provide a deeper understanding of the psychological and behavioural complexities of transitioning out of elite sport and highlight critical gaps in current support systems. Notable limitations include the small sample size from individual sports despite achieving saturation of main themes. Additionally, the requirement for English fluency excluded those from culturally and linguistically diverse backgrounds, meaning their perspectives were not captured. Furthermore, there is potential for selection bias, as individuals with stronger opinions or more challenging athlete retirement transitions may have been more motivated to participate. The use of snowball sampling and personal networks may have also influenced the types of narratives captured, favouring participants who were more connected or motivated to discuss their experiences. Consequently, perspectives from athletes who experienced smoother transitions or who distanced themselves from sporting networks may be underrepresented. While these methods were appropriate for a sensitive topic and a dispersed population, they reduce the transferability of findings and should be considered when interpreting the results.

Despite these limitations, the findings reinforce the need for dietetic intervention throughout an athlete’s career, from the early stages of training through to post-retirement. Current dietary advice for the general population does not sufficiently address the unique psychological or physiological needs or athletes transitioning from elite sport. Athlete dietetic support should consider the long-term influence of sporting culture on eating behaviours and identity to achieve positive health outcomes. Equally important is concurrent psychological support to facilitate behaviour change and help athletes adapt to structural and lifestyle transitions, especially given the fact that loss of identity and body image concerns are well-recognised barriers to healthy adjustment [1].

Culture change within elite sport is also required. Creating environments that promote healthy relationships with food and oneself during an athlete’s career could assist with a healthy post-retirement transition. Several participants noted that they would have performed better if fuelled appropriately, which echoes findings from previous research linking inadequate nutrition to impaired performance and increased risk of injuries [7]. Overall, the experiences of former athletes in this study appeared consistent across the participants in the sample. These observations are exploratory, though, and should not be interpreted as being definite or universally applicable. Additional exploration in more diverse cohorts would be useful. Common challenges included loss of identity, disrupted or loss of routine, dysfunctional eating behaviours, and lingering impacts of sports culture. Over time, the participants developed more balanced, intuitive relationships with food, their body, and their psyche, highlighting the potential for adaptation. 

## 5. Conclusions

The findings of this study reinforce the need for integrated dietetics and psychological support during and beyond an athlete’s career, in addition to cultural shifts within sports. Future research should explore whether digital or artificial intelligence-supported resources can be used to help with successfully transitioning to post-retirement life. Evaluating the long-term outcomes of structured tailored transition programmes, including self-directed tools, could represent an important step for supporting athletes’ wellbeing post-retirement.

## Figures and Tables

**Figure 1 nutrients-17-03920-f001:**
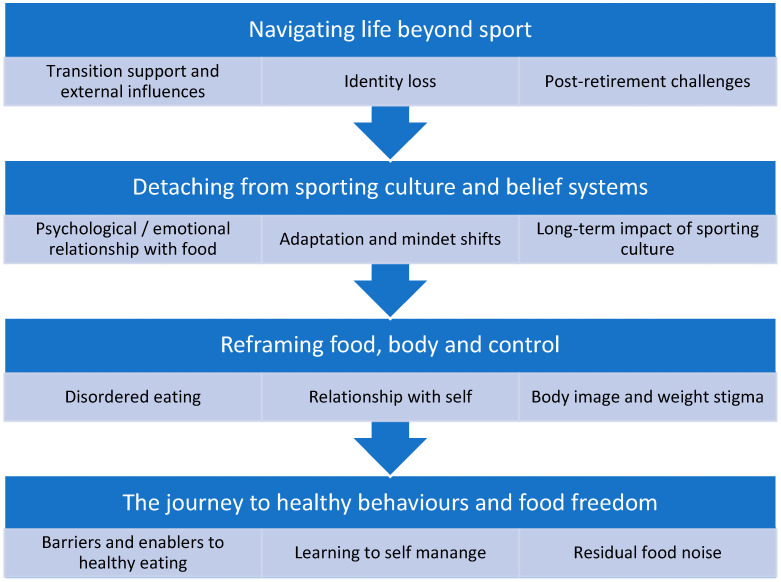
Thematic map of the experiences of elite athletes regarding healthy eating after retirement. The four themes are presented in a sequence to illustrate a common trajectory. However, this arrangement is intended to be illustrative rather than a prescriptive model. Participants’ accounts often reflected cyclical, overlapping, and simultaneous processes that reinforced that identity reconstruction and eating autonomy unfolded in nonlinear ways.

**Table 1 nutrients-17-03920-t001:** Semi-structured interview guide.

Demographic Questions
What sport/s did you compete in at a national and/or international level? How long did you compete at this level? How long ago did you conclude competing at this level? Now I will show you an image that classifies the type of athlete you were. What type were you? (See Appendix A) What is your age, weight, and height now? What was your age, weight, and height (young athletes may have since grown) when competing?
**Interview Questions**
1. What is the first thing that comes to mind when thinking about making food choices after you ceased competitive high-level or elite sport? (Prompts: Did you feel well prepared? Did you know what to eat and how to adjust your food intake to suit your new lifestyle? Did you feel as though you were eating for health or for performance?) 2. Do you think your level of physical activity has changed since you finished competing? (Prompt: Do you think it has greatly or slightly increased/decreased, or stayed the same?) 3. Since you finished competing, have you been provided with or have you sought out any nutritional information, guidance, or assistance? (Prompts: Can you describe this? Who was this? Why this person? Was the advice helpful? What did they suggest? Did/do you follow this advice?) 4. What impact did competing at a high level/elite level of sport have on you or your eating habits? (Prompts: Did you gain weight? Were there any negative consequences? Were there any positive aspects to this phase?) 5. When you think about your eating habits now, how would you describe them? (Prompts: Are there barriers to helping you eat in a healthy way? Are there things that help you? Do you follow a particular diet?) 6. Can you describe any other strategies you have used to help follow a healthy diet? (Prompt: Are there tools you use, e.g., apps to track intake, programmes) 7. When you think about other athletes who have ceased elite competition, what do you think might be the barriers or enablers to following a healthy diet? 8. Do you have any suggestions for how we can better support other athletes with their diet after they cease elite competitive sport? 9. Is there anything else you wish to add? 10. Is there anyone you think we should speak to that may be able to share their experiences?

**Table 2 nutrients-17-03920-t002:** Participant characteristics.

Type of Sport	Sport	Gender	Tier ^1^	Age During Elite Competition (Height, Weight)	Age at Time of Interview (Height, Weight)
Team sports	American football	Male	3	22–23 (165 cm, 68 kg)	25 (165 cm, 85 kg)
	Baseball	Male	3–4	22–24 (173 cm, 72 kg)	38 (173 cm, 80 kg)
Individual sports	Artistic gymnastics	Female	3–4	12–18 (173 cm, 52 kg)	26 (173 cm, 58 kg)
	Artistic gymnastics	Female	3	5–16 (172 cm, 50 kg)	28 (172 cm, 66 kg)
	Ballet	Female	4	10–19 (171 cm, 50 kg)	30 (171 cm, 65 kg)
	Ballet	Female	4	17–27 (178 cm, 62–68 kg)	30 (178 cm, 68 kg)
	Ballet	Female	4	17–19 (157 cm, 38–42 kg)	26 (157 cm, 46 kg)
	Ballet	Male	4	22–25 (183 cm, 78 kg)	26 (183 cm, 90 kg)
	Bodyboarding	Male	4	25–31 (180 cm, 78 kg)	39 (180 cm, 82 kg)
	Cycling	Male	4	16–17 (171 cm, 61 kg)	18 (171 cm, 68 kg)
	Swimming	Female	3	13–16 (165 cm, 58 kg)	23 (167 cm, 65 kg)
	Swimming	Female	3	12–17 (166 cm, 56 kg)	27 (166 cm, 66 kg)
	Triathlon	Female	4	23–28 (163 cm, 54 kg)	41 (163 cm, 54 kg)
	Triathlon	Female	5	17–30 (175 cm, 65 kg)	30 (175 cm, 69 kg)
Defence Force ^2^	Special Operations Command	Male	4	19–23 (173 cm, 80 kg)	25 (173 cm, 72 kg)
	Special Operations Command and Australian Services Rugby League	Male	3	Defence Force: 21–64 (176 cm, 87 kg) Rugby League: 21–30 (176 cm, 87 kg)	65 (176 cm, 99 kg)

^1^ Tier 3: Highly trained/National level; Tier 4 Elite/International level; Tier 5 Olympic Level/World class. See Appendix A for more information. ^2^ Defence Force specialty removed to preserve participant anonymity.

**Table 3 nutrients-17-03920-t003:** Athletes’ descriptions of sporting culture attitudes and behaviours.

Ideal physique	“*And it’s just yeah, the kind of attitudes like, oh, you have to be so lean to swim fast. And it’s like we used to do testing days, like they’d pull us in from, like, all across the state as because we met, like, a certain threshold. They were like, Right girls, shirts off. We’re all doing skin folds now, like we’re 14, like that. That’s not necessary. And then everyone would come out and um, would compare results*” (Participant 11, Swimming) “*So it’s also, like, perpetuated by the teachers in class paying more attention to dancers in smaller bodies or people with, like, thinner physiques*” (Participant 7, Ballet) “*I think being in a sport where you are racing in like, a swimming costume, essentially, swimming, riding and running and then so like, I guess, just like your body’s on display, but also, like, naturally, the smaller and the leaner you are, the faster you go, not necessarily, the healthier you are. But that was like a strong message, like strong message that was sent out for a long time. So I think, like the pressures of, I guess, looking a certain way, not only for like, body image, but also for performance, definitely then had a very negative impact on, like, my relationship with food for a long time. And then even after I kind of realized how negative that was, or, like, how toxic that was for performance and just my overall health, it still probably took another two years to be able to be able to actually turn it around and really change those habits*” (Participant 14, Triathlon) “*[what influenced food choices] Definitely, body image. It was, I think gymnastics, very female dominated sport, and a lot of it is revolved around body image. And, you know, making sure you’re a certain figure when competing you’re up against other female athletes. So there’s a lot of comparison with body image in the sport. So I think that was the biggest thing that led me to look a certain way. But yeah, from a food perspective, it was all diet related foods. It was whatever was in the media at the time [skinny me teas, diet jelly as a meal]. It was all based around, you know, trying to have that perfect physique for a female athlete*” (Participant 4, Artistic Gymnastics)
Weight-biased culture	“*We would get weighed and in front of all of other gymnast as well. So it was that, like we were 12, like we didn’t really think, Oh, well, I’m taller, so naturally I would probably weigh more, like, it’s just, oh, I weigh more than you, and then that changes how much you’re eating, too. So I think that had an influence that had a negative influence on food intake*” (Participant 3, Artistic Gymnastics) “*We would be called into our coach’s office and, like, we were pretty much stripped down right to the leotard and, like, jumped on a scale. And honestly, the coach didn’t even know anything about nutrition or body weight, so it was purely just a numbers game for him. And if it looked high and you like, we looked a bit more muscular than maybe the norm, like we would, like, we would be told we’d have to go sit on the bike for 60 min during training. And we were, like, 12 years old.*” (Participant 3, Artistic Gymnastics)
Food demonisation	“*A lot of [coaches] would just be suggesting to eat less, a lot. A big advice for me was to cut out carbohydrates completely, since carbohydrates were just bloat and it’s not necessary, apparently. And anytime snacking was suggested, it would be something small, like a carrot cut up, or an apple cut up, or something like that.*” (Participant 8, Ballet)
Restrained eating	“*We were told to eat very little, and [coaches] would try to restrict that. The [director] knows two words, and that’s eat little. Eat little. They don’t really care about food groups and how it actually works*” (Participant 10, Cycling) “*We actually weren’t allowed to eat in the gym. We typical gymnastics community of like, being slim, tall, lean, all of that. So we weren’t allowed to eat. We actually weren’t allowed to have any fluids out. And so, it was all being like sneaking food*” (Participant 3, Artistic Gymnastics)
Food misconceptions	“*They don’t really care about food groups and how it actually works. They just [believe], if you eat little food and train hard and go hard, you’ll go fast*” (Participant 10, Cycling)
Guilt associated with eating	“*We [had] to eat in the bathrooms because we can’t get caught eating in front of the directors. Otherwise, they will take the food away from them. I’d say that that has an effect on why people, when they leave the sport, might not want to restrict their diet, because they’ve been forced to out of their own will, or they’ve had to be secretive of eating more.*” (Participant 10, Cycling)

## Data Availability

The original contributions presented in the study are included in the article/Appendix A, further inquiries can be directed to the corresponding author.

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
