# Peer review of "Navigating Nutrition Beyond Elite Sport: A Qualitative Exploration of Experiences After Retirement"

_nutrients, 2025, doi:10.3390/nu17243920_

Round 1

Reviewer 1 Report

Comments and Suggestions for Authors

Retirement from elite sport often presents athletes with a challenging transitional period. Existing literature indicates a lack of comprehensive research and targeted programs addressing the psychosocial dietary behaviors of retired elite athletes. Whillas and colleagues sought to examine the lived experiences of retired athletes in Australia, with a particular focus on sport-related food culture, as well as past and current eating behaviors and food perceptions. A qualitative approach utilizing semi-structured interviews were used to explore these perspectives.

Page 1, line 36: please give an example of “maladaptive eating behaviors.” (e.g. compensatory exercise??)

Page 2 line 55: Remove comma after “qualitative” and place it after “approach.” However, that word “qualitative” is already used in line 54 and again in line 59. Consider rephrasing the last sentence in this paragraph as follows – “This research examines preparedness of athletes to adopt healthy food choices and identifies factors influencing dietary behaviours during the transition to life beyond sport.”

Page 3 line 59: Please add a reference for relativist ontology

Table 1

  • Pre- and post- bodyweight was asked in the interview, but data were not provided or discussed. Was there verification of reported height and/or weight? (i.e., utilizing a scale for weight or height measure?)

Supplementary materials

  • I would consider revising the article’s title, specifically removing the word “elite”. Describing the whole cohort as “elite” seems a bit misleading when 5-7 athletes out of the 16 athletes were classified as tier 3 “highly trained/national level.” Using the Participant Classification Framework figure, I would consider only athletes classified as level 4 and 5 as “elite.”

The use of an oxford comma was not consistent throughout the manuscript. Overall, seems like oxford commas were not used in the Introduction, Materials/Methods sections, while they are used in the Results section. Discussion section is mixed with both. Please revised to be consistent throughout.

Page 7 line 174: correct “from the from the beliefs”

Page 7 line 192: please include clarification on what “it” refers to in participant 3’s quote [add brackets]

Page 10 line 231: can you elaborate on “Social media played a role in normalising diverse athletic bodies and promoting positive self-image.”? Social media is often demonized for spreading poor or inaccurate nutrition information, so I’m curious about the positive role social media played in the athletes’ body-acceptance. Did participants speak on shared-experiences related to anti-food coaches that led to bonding with other athletes online?

Page 10 line 265: correct “One athlete One participant shared…”

Add spaces after sentence before (reference#) on page 11 lines 288 and 308

Author Response

Retirement from elite sport often presents athletes with a challenging transitional period. Existing literature indicates a lack of comprehensive research and targeted programs addressing the psychosocial dietary behaviours of retired elite athletes. Whillas and colleagues sought to examine the lived experiences of retired athletes in Australia, with a particular focus on sport-related food culture, as well as past and current eating behaviors and food perceptions. A qualitative approach utilizing semi-structured interviews were used to explore these perspectives. 

Page 1, line 36: please give an example of “maladaptive eating behaviors.” (e.g. compensatory exercise??)

 This has now been included and states: “increased vulnerability to maladaptive eating behaviours such as restrictive eating, compensatory changes to exercise, as well as…”

Page 2 line 55: Remove comma after “qualitative” and place it after “approach.” However, that word “qualitative” is already used in line 54 and again in line 59. Consider rephrasing the last sentence in this paragraph as follows – “This research examines preparedness of athletes to adopt healthy food choices and identifies factors influencing dietary behaviours during the transition to life beyond sport.”

 This has now been amended as suggested

Page 3 line 59: Please add a reference for relativist ontology

 This is now included (Swift, J.A. and Tischler, V. (2010), Qualitative research in nutrition and dietetics: getting started. Journal of Human Nutrition and Dietetics, 23: 559-566. https://doi.org/10.1111/j.1365-277X.2010.01116.x)

Table 1. Pre- and post- bodyweight was asked in the interview, but data were not provided or discussed. Was there verification of reported height and/or weight? (i.e., utilizing a scale for weight or height measure?)

 The weights and heights of athletes has now been included in Table 2. No verification was conducted to confirm these as they were conducted online not in person.

Supplementary materials

  • I would consider revising the article’s title, specifically removing the word “elite”. Describing the whole cohort as “elite” seems a bit misleading when 5-7 athletes out of the 16 athletes were classified as tier 3 “highly trained/national level.” Using the Participant Classification Framework figure, I would consider only athletes classified as level 4 and 5 as “elite.”

 Using the definitions contained with the Athlete Participant Classification Framework, we have amended wording to better reflect the population. This includes amending the abstract.

The use of an oxford comma was not consistent throughout the manuscript. Overall, seems like oxford commas were not used in the Introduction, Materials/Methods sections, while they are used in the Results section. Discussion section is mixed with both. Please revised to be consistent throughout.

 Punctuation and grammar has been amended to improve this aspect.

Page 7 line 174: correct “from the from the beliefs”

 Now amended

Page 7 line 192: please include clarification on what “it” refers to in participant 3’s quote [add brackets]

 This has been amedned

Page 10 line 231: can you elaborate on “Social media played a role in normalising diverse athletic bodies and promoting positive self-image.”? Social media is often demonized for spreading poor or inaccurate nutrition information, so I’m curious about the positive role social media played in the athletes’ body-acceptance. Did participants speak on shared-experiences related to anti-food coaches that led to bonding with other athletes online?

 This has been amended and one additional quote included to clarify the meaning. “Social media played a role in normalizing healthy eating and food behaviours and promoting positive self-image: “ex Olympians obviously have a much greater platform because of social media, and there's been a real change in attitudes and calling out unhealthy patternsfood is like a social thing, and can be like celebrated and you should enjoy it and it tastes good” (Participant 11, Swimming) .”

Page 10 line 265: correct “One athlete One participant shared…”

 Now amended

Add spaces after sentence before (reference#) on page 11 lines 288 and 308

Now amended

Reviewer 2 Report

Comments and Suggestions for Authors

The manuscript presents a timely qualitative exploration of how former elite athletes adjust their eating behaviours, beliefs, and identity after finishing high-performance sport. The interviews offer rich, emotionally honest material, and several quotations capture the complex psychological shifts that occur during this transition. I appreciate the authors’ intention to centre lived experience and to document aspects of retirement that are often overlooked in sport nutrition research. Despite these strengths, several conceptual and methodological issues weakened the analysis.

  1. Conceptual and Theoretical Clarity

While reading the Introduction, I sought a clear conceptual framework to help me understand how the authors interpret identity disruption, food-related anxiety, and post-sport behavioural change. The narrative introduces many relevant ideas but does not explain how they fit together or what theoretical lens guides the analysis. The absence of defined constructs—particularly “healthy eating,” “food freedom,” and “nutritious diet”—made it difficult to see how the authors distinguished participants’ subjective meanings from normative assumptions.

A more precise articulation of the conceptual boundaries would substantially strengthen the manuscript and help situate the study within broader work on athlete identity transitions and food psychology.

  1. Methodological Transparency and Reflexivity

Sampling and recruitment.

Snowball sampling and personal networks were the primary recruitment strategies. Given the sensitive nature of the topic, I expected some reflection on how these methods may have shaped the sample’s composition or the kinds of narratives that emerged. The manuscript reports the procedure but does not consider potential implications.

Inclusion of defense force personnel

The inclusion of former defense force members was unexpected. The manuscript briefly mentions “regimented dietary routines” as a justification, but I remained unsure whether these experiences align closely enough with elite sport culture to warrant combining them analytically. Their presence raises conceptual questions that deserve clarification.

Analytic depth and reflexivity

The analytic process is described mainly in procedural terms. I was hoping for a more reflective account of how coding decisions were negotiated, how themes were constructed, and how the research team’s perspectives may have shaped the interpretations. With a topic that touches on body image, guilt, and psychological vulnerability, reflexivity becomes essential to ensuring analytic integrity.

  1. Results and Interpretation

Sequential thematic map

The thematic map presents the four themes as sequential. In reviewing the participant quotations, I did not see strong evidence for a linear progression. Many narratives appear cyclical, overlapping, or simultaneous. The sequential framing risks oversimplifying what seems to be a fluid, nonlinear psychological process.

Balance between quotation and analysis

Several quotations are powerful and insightful. Still, in several sections the analysis steps back and allows extended excerpts to stand without sufficient interpretive commentary. I would have appreciated a stronger analytical voice connecting participants’ words to the broader claims of each theme.

Claims of thematic consistency

The manuscript repeatedly states that experiences were similar across gender, age, and sport discipline. With a sample of 16 participants and no explicit comparative analysis, these claims feel too strong. More cautious phrasing would avoid overstating generalizability.

  1. Discussion and Literature Integration

The Discussion engages with relevant literature but tends to rely on confirmatory studies. More critical engagement—highlighting how the findings extend, challenge, or complicate existing knowledge—would strengthen the manuscript's scholarly contribution. The brief reference to AI tools such as ChatGPT is interesting, but it requires clearer theoretical grounding to avoid appearing anecdotal.

  1. Ethical Considerations

Many participants describe emotionally charged experiences involving food guilt, body scrutiny, and identity loss. The Methods section acknowledges ethics approval, but the manuscript does not reflect on emotional safety, interviewer–participant dynamics, or strategies for handling distress. Greater attention to ethical reflexivity would strengthen the study.

  1. Writing and Structure

The manuscript is generally well written, but I noticed minor issues that affect precision:

  • grammatical inaccuracies (“from the from the beliefs,” p. 7),
  • vague or undefined terminology,
  • occasional blending of results with interpretation,
  • some repetition within thematic descriptions.

These issues are easily fixable.

Author Response

The manuscript presents a timely qualitative exploration of how former elite athletes adjust their eating behaviours, beliefs, and identity after finishing high-performance sport. The interviews offer rich, emotionally honest material, and several quotations capture the complex psychological shifts that occur during this transition. I appreciate the authors’ intention to centre lived experience and to document aspects of retirement that are often overlooked in sport nutrition research. Despite these strengths, several conceptual and methodological issues weakened the analysis.

  1. Conceptual and Theoretical Clarity

While reading the Introduction, I sought a clear conceptual framework to help me understand how the authors interpret identity disruption, food-related anxiety, and post-sport behavioural change. The narrative introduces many relevant ideas but does not explain how they fit together or what theoretical lens guides the analysis. The absence of defined constructs—particularly “healthy eating,” “food freedom,” and “nutritious diet”—made it difficult to see how the authors distinguished participants’ subjective meanings from normative assumptions.

A more precise articulation of the conceptual boundaries would substantially strengthen the manuscript and help situate the study within broader work on athlete identity transitions and food psychology.

The manuscript includes reference to the use of a relativist onological lens on page 2. Additional wording has been included in the introduction to articulate the conceptual boundaries.

  1. Methodological Transparency and Reflexivity

Sampling and recruitment.

Snowball sampling and personal networks were the primary recruitment strategies. Given the sensitive nature of the topic, I expected some reflection on how these methods may have shaped the sample’s composition or the kinds of narratives that emerged. The manuscript reports the procedure but does not consider potential implications.

We have added a reflexive statement acknowledging how snowball sampling and reliance on personal networks may have influenced the composition of our sample and the narratives captured. Specifically, these strategies likely favoured participants who were more willing to share their experiences or who maintained connections within sporting communities, which may have amplified certain perspectives (such as those with strong opinions about transition challenges). We note this as a limitation and have clarified that while these methods facilitated access to a hard-to-reach population, they may reduce transferability of findings. This reflection has been incorporated into the revised Methods and limitations sections.

Inclusion of defense force personnel

The inclusion of former defense force members was unexpected. The manuscript briefly mentions “regimented dietary routines” as a justification, but I remained unsure whether these experiences align closely enough with elite sport culture to warrant combining them analytically. Their presence raises conceptual questions that deserve clarification.

Thank you for raising this point. We have clarified the rationale for including former defence force personnel and how their experiences align conceptually with elite sport culture. These individuals were selected because their training environments share key characteristics with elite sport: regimented dietary routines, strict body composition standards, and performance-driven physical conditioning. Like athletes, they operate under external control of food intake, experience identity shifts upon leaving service, and face challenges in recalibrating eating behaviours when structured routines end. Including this group allowed us to explore whether these shared cultural and psychological dynamics produce similar post-transition experiences. We have added text to the Methods and Discussion sections to explain this alignment and note that while their narratives were analysed alongside athletes, we acknowledge this as a conceptual boundary and interpret findings with caution

Analytic depth and reflexivity

The analytic process is described mainly in procedural terms. I was hoping for a more reflective account of how coding decisions were negotiated, how themes were constructed, and how the research team’s perspectives may have shaped the interpretations. With a topic that touches on body image, guilt, and psychological vulnerability, reflexivity becomes essential to ensuring analytic integrity.

The methods has now been amended to better reflect the use of reflexivity and expanded the description of the analytic process. We hope this is improved for the reviewer.

  1. Results and Interpretation

Sequential thematic map

The thematic map presents the four themes as sequential. In reviewing the participant quotations, I did not see strong evidence for a linear progression. Many narratives appear cyclical, overlapping, or simultaneous. The sequential framing risks oversimplifying what seems to be a fluid, nonlinear psychological process.

This has now been expanded in the accompany text below the figure to state: Thematic map of the experiences of elite athletes regarding healthy eating after retirement. The four themes are presented in a sequence to illustrate a common trajectory. However, this arrangement is intended as illustrative rather than prescriptive model. Participants’ accounts often reflected cyclical, overlapping, and simultaneous processes, that reinforced that identity reconstruction and eating autonomy unfolded in non linear ways.

Balance between quotation and analysis

Several quotations are powerful and insightful. Still, in several sections the analysis steps back and allows extended excerpts to stand without sufficient interpretive commentary. I would have appreciated a stronger analytical voice connecting participants’ words to the broader claims of each theme.

Additional wording is now included throughout to link analysis with interpretation. We hope this improves the manuscript.

Claims of thematic consistency

The manuscript repeatedly states that experiences were similar across gender, age, and sport discipline. With a sample of 16 participants and no explicit comparative analysis, these claims feel too strong. More cautious phrasing would avoid overstating generalizability.

Thank you for noting this concern. We have revised the language throughout the manuscript to avoid overstating generalisability.

  1. Discussion and Literature Integration

The Discussion engages with relevant literature but tends to rely on confirmatory studies. More critical engagement—highlighting how the findings extend, challenge, or complicate existing knowledge—would strengthen the manuscript's scholarly contribution. The brief reference to AI tools such as ChatGPT is interesting, but it requires clearer theoretical grounding to avoid appearing anecdotal.

We have revised the wording to be clearer in this aspect.it now reads: “Several participants described using podcasts and generative AI tools like ChatGPT during the transition. These tools can operate as autonomy supporting resources and connect users to non judgmental support and information. Risks include accuracy, accountability, privacy and over reliance without clinical oversight. . These approaches are largely absent form the current literature, suggesting early signals of autonomy oriented self care that may reduce pressure during the transition period (7). Importantly, athletes emphasized the need for time away from sport before engaging with structured programs, indicating that initial support should be external to sporting organisations to foster trust and uptake. Developing sport-specific digital resources, such as transition-focused podcasts or educational modules on safe and ethical AI use for nutrition may bridge this gap until athletes are ready for formal services like tailored webinars, peer networks, and psychological counselling. This concept of self-directed digital support remains undocumented in prior literature and represents a promising research avenue as an enabler of athlete centred dietetic and psychological care.”

  1. Ethical Considerations

Many participants describe emotionally charged experiences involving food guilt, body scrutiny, and identity loss. The Methods section acknowledges ethics approval, but the manuscript does not reflect on emotional safety, interviewer–participant dynamics, or strategies for handling distress. Greater attention to ethical reflexivity would strengthen the study.

Thank you for noting this. In response, both senior researchers are qualified health professionals with extensive counselling experience and were available to provide debriefing if desired. In cases of severe distress, mental health first aid protocols were in place, including adopting an attitude of acceptance and empathy, active listening, and positive body language. Mental first aid or apparent distress was not required or evident for any participant

  1. Writing and Structure

The manuscript is generally well written, but I noticed minor issues that affect precision:

  • grammatical inaccuracies (“from the from the beliefs,” p. 7),
  • vague or undefined terminology,
  • occasional blending of results with interpretation,
  • some repetition within thematic descriptions.

These issues are easily fixable.

These minor issues have been attended to where relevant